# Global distribution and conservation status of ecologically rare mammal and bird species

Nicolas Loiseau [1,2,3,9✉], Nicolas Mouquet [1,4,9✉], Nicolas Casajus [4], Matthias Grenié[3], Maya Guéguen[2], Brian Maitner [5], David Mouillot [1,6], Annette Ostling[7], Julien Renaud[2], Caroline Tucker [8], Laure Velez[1], Wilfried Thuiller[2,10] & Cyrille Violle[3,10]

Identifying species that are both geographically restricted and functionally distinct, i.e. supporting rare traits and functions, is of prime importance given their risk of extinction and their potential contribution to ecosystem functioning. We use global species distributions and functional traits for birds and mammals to identify the ecologically rare species, understand their characteristics, and identify hotspots. We find that ecologically rare species are disproportionately represented in IUCN threatened categories, insufficiently covered by protected areas, and for some of them sensitive to current and future threats. While they are more abundant overall in countries with a low human development index, some countries with high human development index are also hotspots of ecological rarity, suggesting transboundary responsibility for their conservation. Altogether, these results state that more conservation emphasis should be given to ecological rarity given future environmental conditions and the need to sustain multiple ecosystem processes in the long-term.

[1] MARBEC, Univ Montpellier, CNRS, Ifremer, IRD, 34 095 MONTPELLIER, Cedex 5, France. [2] Univ. Grenoble Alpes, CNRS, Univ. Savoie Mont Blanc, LECA, Laboratoire d'Ecologie Alpine, F-38000 Grenoble, France. [3] CEFE UMR 5175, Univ Montpellier—CNRS—EPHE—IRD—Univ. Paul-Valéry 3, Montpellier, France. [4] FRB—CESAB, Institut Bouisson Bertrand. 5, rue de l'École de médecine, 34000 Montpellier, France. [5] Department of Ecology and Evolutionary Biology, University of Arizona, Tucson, AZ 85721, USA. [6] Institut Universitaire de France, IUF, Paris 75231, France. [7] Ecology and Evolutionary Biology, University of Michigan, Ann Arbor, MI 48109, USA. [8] Environment, Ecology and Energy Program, University of North Carolina at Chapel Hill, 3202 Murray/Venable Hall, CB#3275, Chapel Hill, NC 27599-3280, USA. [9] These authors contributed equally: Nicolas Loiseau, Nicolas Mouquet. [10] These authors jointly supervised this work: Wilfried Thuiller, Cyrille Violle. ✉email: nicolas.loiseau1@gmail.com; nicolas.mouquet@cnrs.fr

Most species are geographically restricted, and rarity has become one of the cornerstones of many ecological studies and conservation strategies for decades[1–4]. Rare species are of particular concern because they tend to have a high extinction risk[5] and may even face a "double jeopardy"[6] as low abundance species are also often narrowly distributed. While it has long been assumed that rare species should weakly contribute to ecosystem functioning and functional diversity, recent studies have challenged this belief. Rare species may indeed contribute disproportionately to the diversity of traits within a community or a region[7–9]. Since high trait diversity is usually assumed to enhance ecosystem functioning, these rare species thus support unique traits or functions that might be irreplaceable[10,11]. In other words, rarity does not only relate to the mere abundance or geographic extent of species but also to their functional distinctiveness[10,11]. If rare species are not redundant with other species and instead hold unique combinations of traits, they will likely contribute disproportionately to ecosystem functioning and associated services[12]. These two facets of rarity have been widely discussed in the literature[1–4,13,14] but they have rarely been combined so far[15]. In the context of the ever-increasing biodiversity crisis[16] and environmental uncertainty, it is now fundamental to understand the ecological characteristics of species that are both geographically restricted and functionally distinct, to map their distribution and predict their vulnerability to current and future threats.

Which species are facing ongoing and future threats the most? It has indeed been shown that species with restricted geographic distributions have the highest risk of extinction under most future climate scenarios[17]. Being at risk of extinction in the face of global changes and not targeted by current conservation programs would be the worst-case scenario for ecologically rare species and the unique functions they support. In any case, the precautionary principle requires a sound assessment of the global patterns of ecological rarity.

For a global assessment of ecological rarity, it is necessary to build the functional space of species worldwide based on their relative position in the Eltonian niche space[18–20] also known as the trophic niche space, which focuses on traits related to biotic interactions and resource–consumer dynamics[21]. To that end, worldwide data on Eltonian traits, including the characterization of diet and foraging activity, are relevant candidate features[22]. Global analyses on mammals and birds[20] indeed led to interesting findings about the shape of the functional space of these taxa and about the global distribution of their functional diversity, notably identifying hotspots of functional redundancy[23–26]. They also highlighted a spatial mismatch between protected areas and taxonomic, phylogenetic, and functional diversity worldwide[27]. Recent findings also showed that threatened birds and mammals are more functionally distinct than non-threatened species[14]. Taken together these results call for a global assessment of both facets of rarity, which will be facilitated by the development of a theoretical corpus and associated methodology[13,15].

Here, we defined and assessed the ecological rarity of mammals (4654 species) and birds (9287 species), two threatened vertebrate clades that are largely distributed across the globe and support important ecological functions in ecosystems such as seed dispersal, trophic interaction, and nutrient cycling[25,26]. For each species, we estimated ecological rarity as a combination of geographical restrictiveness and functional distinctiveness. Geographical restrictiveness is simply defined as the inverse of geographic distribution while functional distinctiveness is based on functional trait dissimilarity (i.e. the extent to which the traits of a given species are distinct compared to all the other species). Combining this information could have been achieved by using an additive or multiplicative framework[13] but this strategy might

lead to blurred and hard-to-interpret information: species with very high restrictiveness and low distinctiveness or with very low restrictiveness and high distinctiveness would display the same level of ecological rarity. To overcome this limitation, we used the intersection of both distributions: we thus defined ecologically rare and common species as those having values of both functional distinctiveness and geographical restrictiveness higher than 75% or lower than 25% of the entire species pool, respectively (Fig. 1). Using this combined information, we investigated (i) which are the ecologically rare species and how they are distributed across functional space and the tree of life?; (ii) how are they geographically distributed and do they follow general biodiversity patterns?; and finally, (iii) what current and future threats are they facing in the Anthropocene and are they covered by current protection areas?

## Results

**Global functional and phylogenetic characteristics of ecologically rare species.** We obtained two distributions based on (1) the geographical range of species compared to the geographic extent of all species in the global pool (Fig. 1) and (2) the functional distinctiveness of species trait values relative to the other species of the global pool (the average functional distance of a species to all the others). 237 mammals (5%) and 573 birds (6%) were defined as ecologically rare (i.e. both functionally and geographically), while 200 (4%) and 569 (6%) were ecologically common.

We projected species into a global functional space based on their traits (see "Methods"). For both taxa, we found that the first axis (PC1) was not correlated to functional distinctiveness which was mostly explained by the second and third axes for mammals and by the second and fourth axes for birds (see Supplementary Figs. 1 and 2). Ecologically rare mammals were mainly nocturnal frugivores with relatively large body sizes (e.g., bats, lemurs) or small invertebrate-eaters (e.g., some rodents, bats, or Eulipotyphla). Ecologically rare birds were mainly frugivores or nectarivores (e.g., hummingbirds) or diurnal piscivores (e.g. large marine birds). Despite their small number, ecologically rare species occupied a large volume in the functional space for both taxa, and did not overlap with ecologically common species, which fill a much smaller volume of the functional space (Fig. 2, see Supplementary Table 1 and Table 2). In case distinctiveness was primarily driven by one or two traits, we would expect ecologically rare species to occupy less functional space than common and average species. Here, ecological rarity is widespread across the functional space.

We plotted ecological rarity on the tree of life and tested phylogenetic signal of ecological rarity using the D index[28]. We found that closely related species were not necessarily more similar in their degree of ecological rarity than distantly related species (Fig. 3, $D = 0.53$ and $D = 0.56$, respectively). However, we found a phylogenetic clustering of ecological rarity for several orders. The orders with the highest concentration of ecologically rare species were the Primates and Chiroptera for mammals and Psittaciformes, Caprimulgiformes and Apodiformes (Strisores) for birds. Finally, we found that ecologically rare species were on average no more evolutionary distinct than other species (see Supplementary Fig. 3).

**Global distribution of ecological rarity.** To identify the geographical 'hotspots' of ecological rarity, we mapped the number of ecologically rare species within each 50 by 50 km cell around the world and examined its spatial congruence with ecological commonness and total species richness. Ecological rarity was aggregated on only 2.8% and 8.9% of the total grid cells for

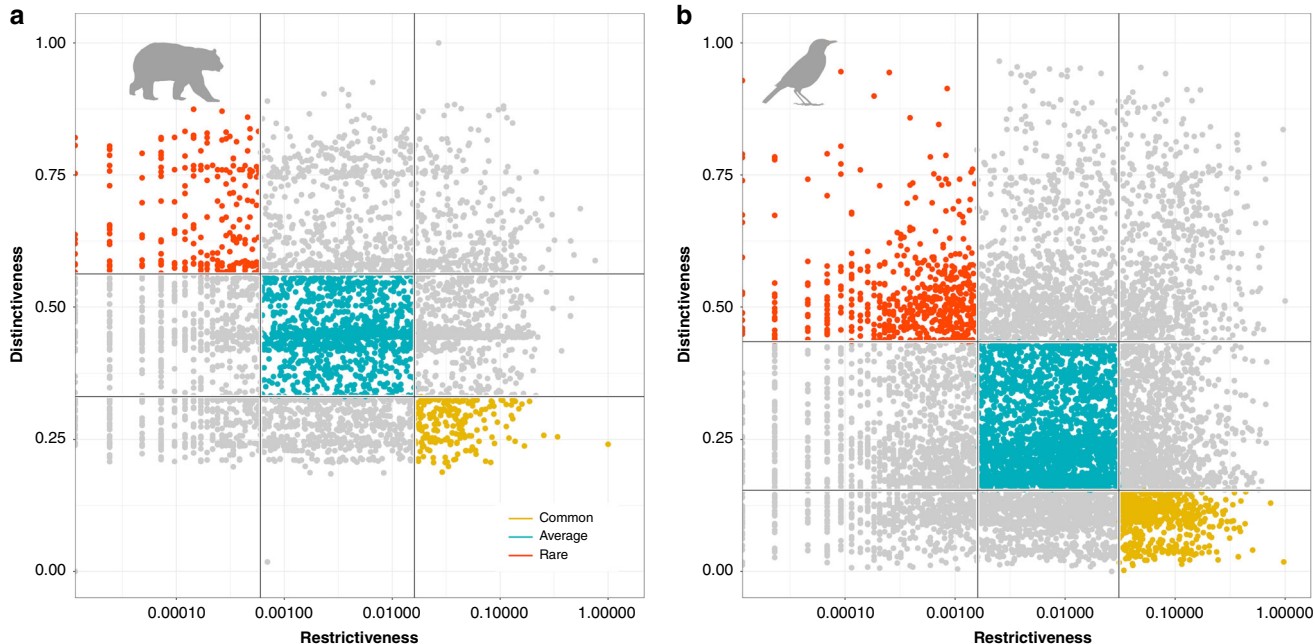

**Fig. 1 Ecologically rare species are defined by the combination of geographical restrictiveness and functional distinctiveness. a, b** Relationship between functional and geographical restrictiveness of mammals and birds. Geographical restrictiveness is defined as the inverse of geographic distribution while functional distinctiveness measures how the traits of a species are original compared to all the other species. We use log(1- restrictiveness) for clarity as most of the restrictiveness values are close to 0 for mammals and birds). Each species is represented by a dot, ecologically rare species are in red, average in blue, and common in orange. Icons were generated using R (*rphylopic* package) and are under the Public Domain Dedication 1.0 license.

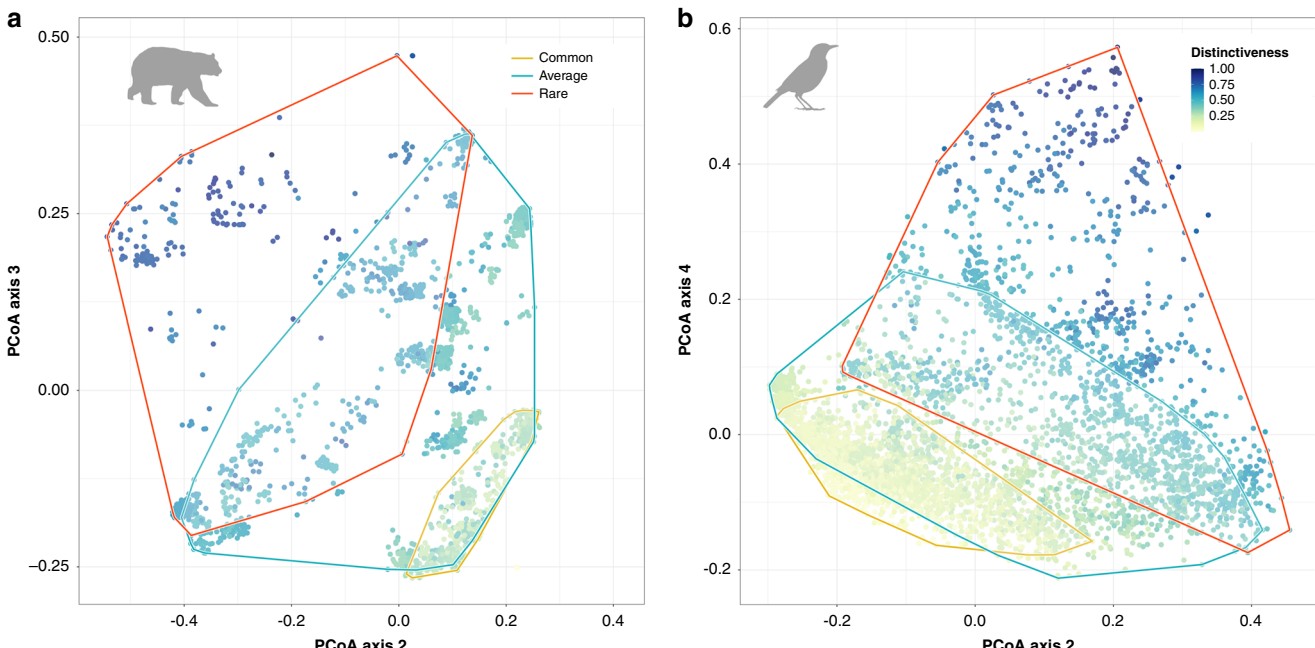

**Fig. 2 Ecologically rare fill a wide breadth of ecological strategies in the Eltonian niche space.** Principal Coordinates Analysis (PCoA) representing the functional space of mammals (**a**) and birds (**b**) based on their functional traits. The two axes (PC2 and PC3 for mammals, PC2 and PC4 for birds,) explain respectively 35.3% and 26.4% of the variance in the distance matrix (see Supplementary Tables 1 and 2). Species are colored by their functional distinctiveness values. Functional space of ecologically rare species is in red, common, and average species are respectively in orange and blue. Icons were generated using R (*rphylopic* package) and are under a public the Public Domain Dedication 1.0 license.

mammals and birds, with maximum values of 12 and 28 species, respectively. Altogether, only 1.1% of the global land hosted at least one ecologically rare species of both taxa. Ecological rarity of mammals predominantly occurred in the tropics and in the Southern Hemisphere, peaking in Indonesian islands, Madagascar, and Costa Rica (Fig. 4). Ecological rarity of birds predominantly occurred in mountainous tropical and subtropical regions, peaking in New Guinea, Indonesia, the Andes, and

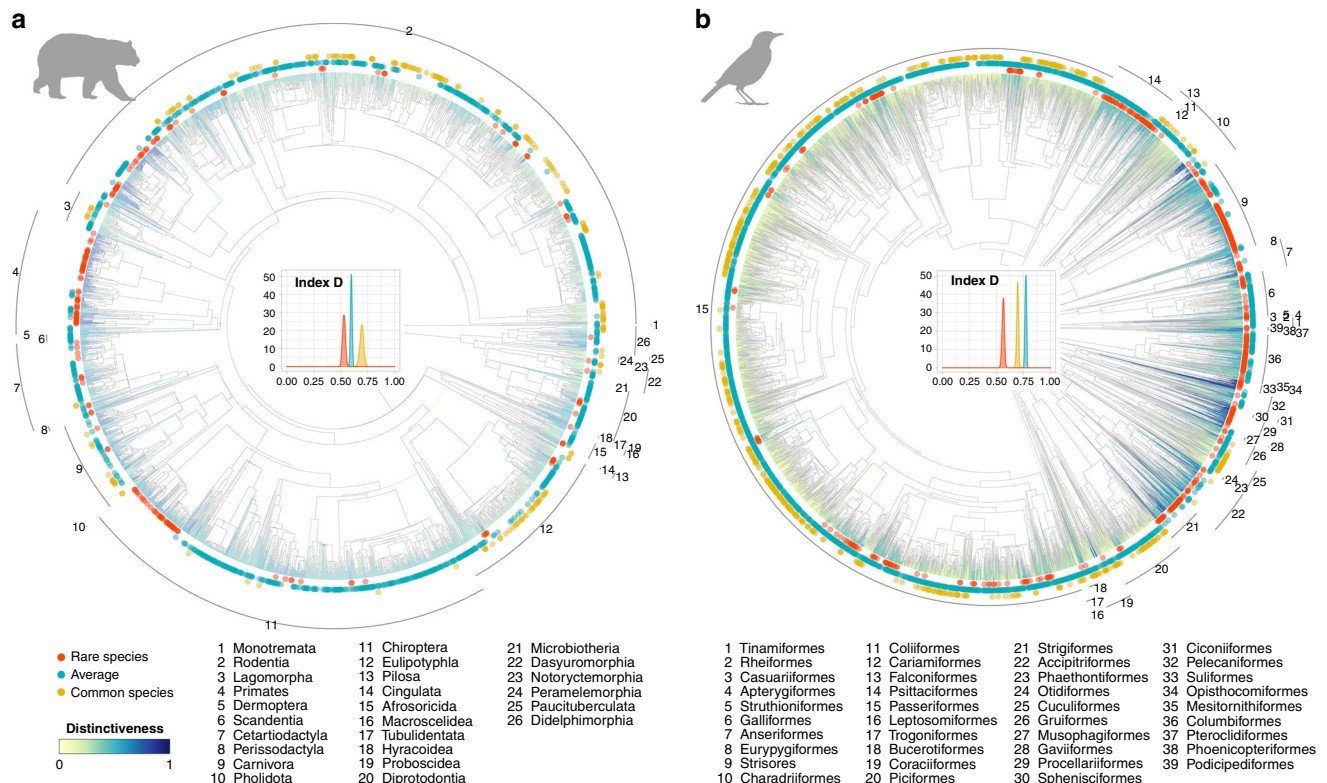

**Fig. 3 Closely related species were not necessarily more similar in their degree of ecological rarity than distantly related species but ecological rarity is over-represented in some orders: Primate and Chiroptera for mammals and Psittaciformes, Caprimulgiformes, and Apodiformes (Strisores) for birds. a, b** Phylogenies of terrestrial mammals and birds. Functional distinctiveness values are represented on terminal branches (color gradient). Ecologically rare species are highlighted by red dots, common by orange and average by blue. Distribution of the values of phylogenetic signal of ecological rarities (index D) computed on 100 trees are plotted in the center of the tree. Orders names are indicated on the outside arcs. The figure represents a single phylogeny from the 100 phylogenies generated (see "Methods"). Icons were generated using R (*rphylopic* package) and are under the Public Domain Dedication 1.0 license.

Central America (Fig. 4). Ecological rarity was over-represented on islands with 39% and 30% of grid cells containing ecologically rare mammals and birds. We also found a strong and significant mismatch between the geographical distribution of ecological rarity and commonness for both taxa (Fig. 4, Pearson correlation corrected for spatial autocorrelation[29]: $R^2 = 0.026$, $F = 1.03$, $P = 0.31$, $n = 61,618$ for mammals and $R^2 = 0.012$, $F = 0.05$, $P = 0.82$, $n = 61,618$ for birds). We found a strong congruence between species richness and ecological commonness and a mismatch with ecological rarity, which is agreement with the general finding that most global species richness patterns result from the distributions of the most widespread species[30].

To better test the link between the number of species and the number of ecologically rare species per cell, we simulated this link under the null expectation that ecologically rare species are randomly distributed among cells regardless of the number of species within cells (see "Methods"). We expected more ecologically rare species in species-rich areas. For both taxa, we found that standardized effect size (SES) was higher than expected for all cells hosting at least one ecologically rare species (see Supplementary Figs. 4 and 5), highlighting that these cells host more ecologically rare species than expected by chance independently of the overall species richness within cells. Interestingly, the number of cells with a high value of SES is very low suggesting that a few very particular environments could favor the emergence and maintenance of ecological rarity.

**Ecological rarity under global threats.** We classified species according to their IUCN status. We found that ecologically rare species were disproportionately packed in IUCN threatened categories for both mammals and birds and significantly more threatened than ecologically common species (71% and 44.2% against 2% and 0.5%, respectively, Fig. 5, $P < 0.001$). As expected, geographical restrictiveness, one of the main IUCN criteria to estimate vulnerability, is higher for threatened species (see Supplementary Figs. 6 and 7). However, we also found that threatened species were functionally more distinct (see Supplementary Figs. 6 and 7). A significant proportion of ecologically rare species were also considered as least concerned (13% for mammals and 52% for birds) or non-evaluated (16% for mammals and 3.8% for birds) by the IUCN.

For each ecologically rare species we evaluated exposure to human footprint, human development (HDI) and the number of conflicts, known to influence conservation outcomes[31]. We found that geographical ranges of ecologically rare mammals and birds were respectively $1.35 \pm 1$ and $1.2 \pm 1$ times more overlapped by human footprint than ecologically common species (Fig. 5, $P < 0.001$, see Supplementary Table 3). Ecologically rare mammals occurred in countries with a lower HDI than ecologically common species (Fig. 5, $P = 0.0032$, see Supplementary Table 3). Ecologically rare birds occurred in countries with HDI similar to ecologically common species (Fig. 5, $P = 0.72$, see Supplementary Table 3). Ecologically rare mammals and birds occurred in countries with a number of conflicts not different from common species (see Supplementary Fig. 8, $P > 0.05$, and Supplementary Table 3). However, some countries with a high number of conflicts (e.g. Colombia, Indonesia) host at least five ecologically rare mammals. Several countries (e.g. Philippines) with a low

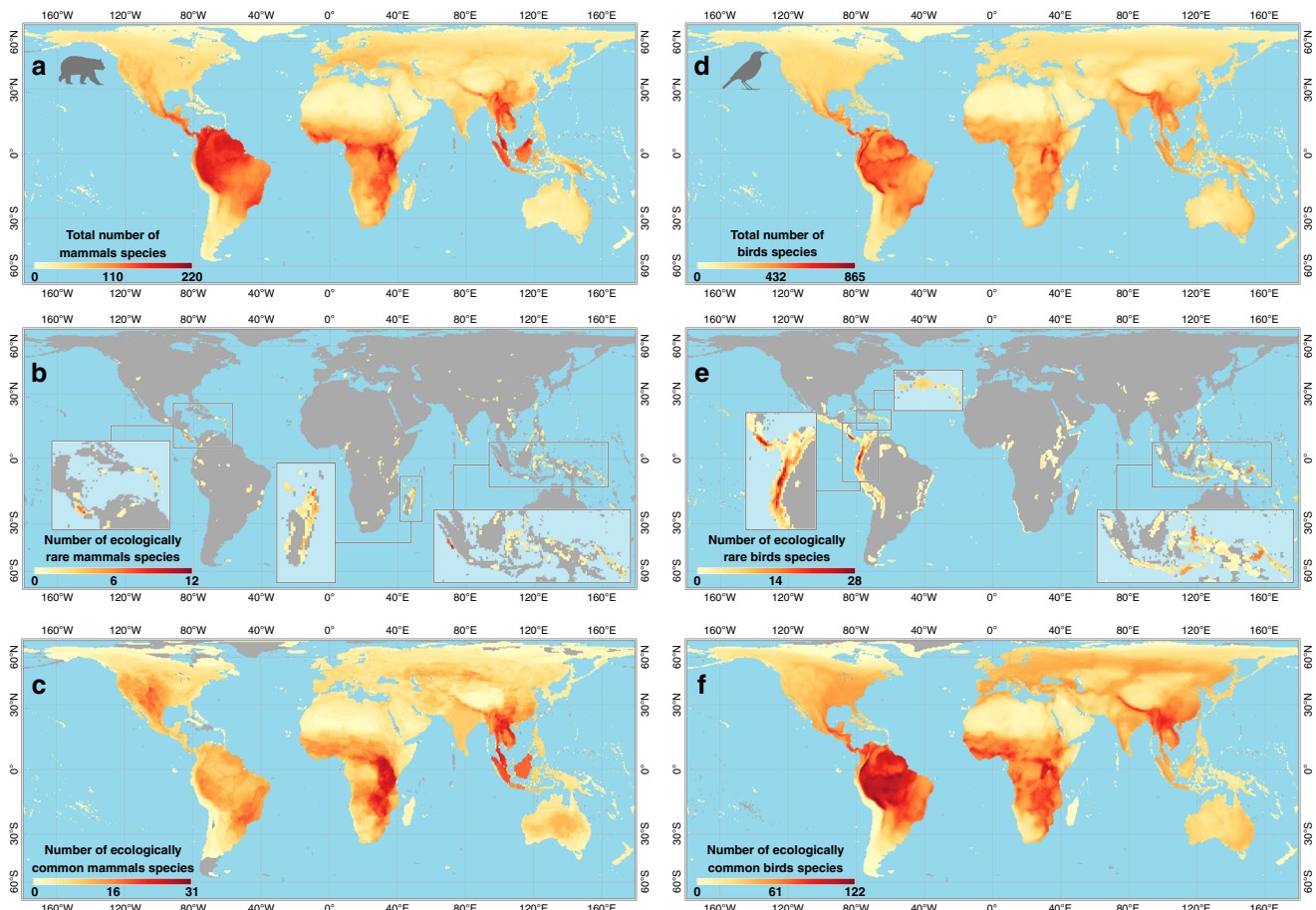

**Fig. 4 Global distribution of mammal and bird species considering only unglaciated areas. a–c** Distribution of total species richness, number of ecologically rare and common mammals, and **d-f** distribution of total species richness, number of ecologically rare and common birds. Icons were generated using R (*rphylopic* package) and are under the Public Domain Dedication 1.0 license.

HDI and a high number of conflicts are hotspots of ecological rarity (19 and 15 ecologically rare mammals and birds, see Supplementary Figs. 9 and 10). On the other hand, some countries with high HDI and low number of conflicts are also hotspots of ecological rarity (e.g. Australia, respectively hosted 5 ecologically rare mammal and 10 bird species).

We then quantified the influence of climate change on ecologically rare and common species. We modeled the current and future climatic suitability for birds and mammals at the global scale (Fig. 5). First, using four species distribution models (SDM), distribution of species was modeled as a function of current climate. Then, we projected their future climatic suitability following RCPs scenarios (see "Methods" and Thuiller et al. [32]). We selected only SDMs that reached high predictive accuracies (TSS > 0.8). Consequently, we kept SDMs outputs for 28% (67 species) and 59% (337 species) of ecologically rare mammal and bird species, respectively. We found winners and losers under future climates for both ecologically rare mammals and birds (Fig. 5). By the time horizon 2041–2060, 36% and 58% of modeled ecologically rare mammals and birds are projected to lose suitable areas (45% and 64% by horizon 2061-2080, see Supplementary Fig. 11). Overall the ecologically rare birds will be more impacted by climate change than common and average ones (Fig. 5, $P = 2.4\mathrm{e}{-}03$, See Supplementary Table 3), but ecologically rare mammals will be less threatened than common and average species (Fig. 5, $P = 3.5\mathrm{e}{-}04$, See Supplementary Table 3).

Finally, to evaluate the potential benefits of conservation efforts on ecological rarity we estimated species-specific target achievement defined as the proportion of geographic ranges covered by protected areas. These specific targets were related to species range sizes with the most restricted species needing more coverage (e.g., 100%) than widespread one (e.g., 10%) to avoid extinction. We found that target conservation achievement of ecologically rare species was lower than for common species for both mammals and birds (Fig. 5, $P < 0.001$, see Supplementary Table 3). Average target achievement for mammals and birds were respectively 15% and 14% for ecologically rare species compared to 31% and 36% for ecologically common species.

## Discussion

We find that both mammals and birds that are ecologically rare fill a much wider breadth of ecological strategies in the Eltonian niche space than ecologically common species. Geographically and locally rare species usually bear distinct traits that could put ecosystem functioning at risk if they go extinct[7,33]. Our findings extend this result as we find that the portion of the functional space filled by ecologically rare species does not overlap much that filled by common species, highlighting a functional complementarity, instead of redundancy, between ecologically rare and common species[34] for both mammals and birds. In particular, we show that specific sets of traits were over-contributing to ecological rarity, echoing the results of Barnagaud et al.[25]

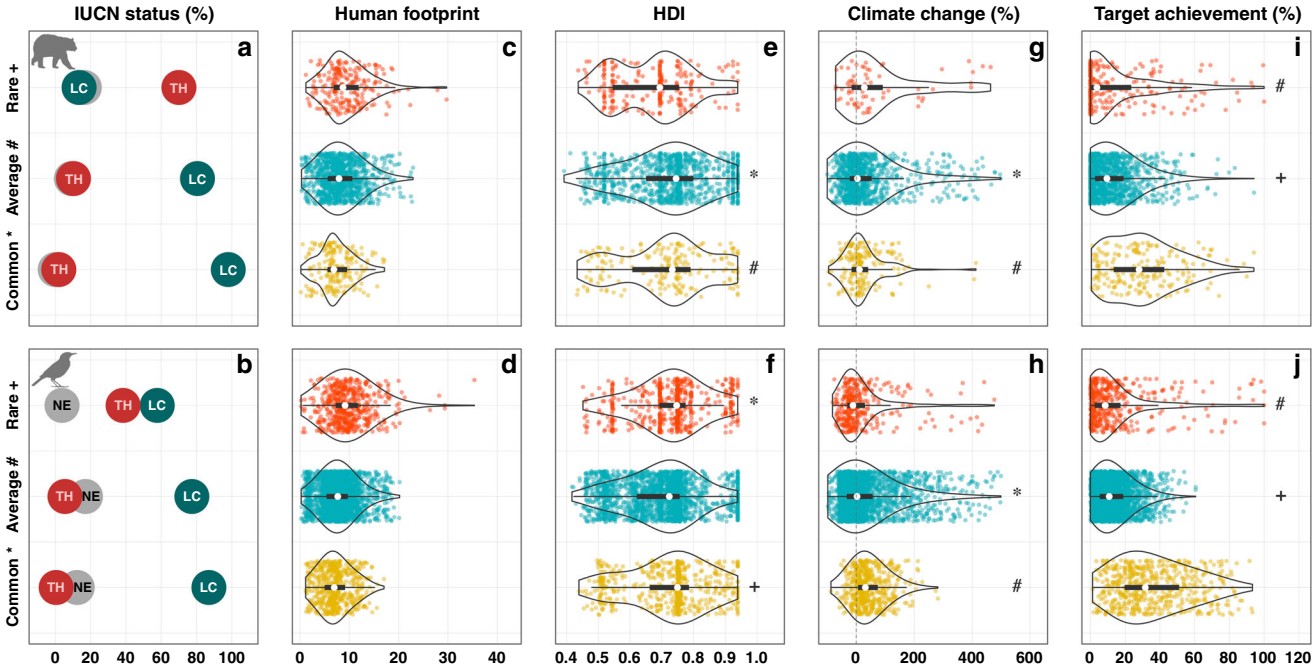

**Fig. 5 Vulnerability to current and future threats of ecologically rare, average, and common mammal and bird species. a, b** Percentage of species per IUCN status according to criteria by the IUCN Red List, Threatened (TH red: Vulnerable, Endangered, Critically Endangered), Least concern (LC green: Least Concern and Near Threatened) and Not Evaluated (NE gray: Not Evaluated or inadequate information). **c, d** Human Footprint that measure the cumulative impact of direct human pressures; **e, f** Human development index (HDI) which is a statistic composite index of life expectancy, education, and per capita income indicators; **g, h** Change in distribution range based on climate change projections (scenario RCP 8.5, Horizon 2041–2060, see Supplementary Fig. 10 for Horizon 2061–2080) and **i, j** Target achievement (extent to which species are represented within PAs regarding their restrictiveness). Symbol indicate significant similar distribution between groups ($P < 0.05$, See Supplementary Table 3 for all p value) via one-way ANOVA and Tukey's post-hoc tests. Icons were generated using R (*rphylopic* package) and are under the Public Domain Dedication 1.0 license.

who found that nectarivory, carnivory, and piscivory are rare diets in birds. A significant proportion of ecologically rare mammals and birds are indeed nectarivores or frugivores. These traits are important for ecosystem functioning as they are directly linked to pollination and seed dispersal, essential for plant survival[35,36] and community stability[37]. For instance, ecologically rare lemurs in Madagascar (such as *Eulemur macaco*[38]) display complex relationships with trees producing large seeds. The extinction of these species would likely amplify the ongoing decline of some tree species under other accelerating global changes in the region[39]. Similarly, while most hummingbirds are ecologically rare, they are the main pollinators of ~7000 plant species and thus maintain ecological networks and crucial ecosystem processes[40]. We also highlight that several seabirds are ecologically rare and play unique functions. By feeding in the open ocean these species transport large quantities of nutrients onto islands, so can enhance the productivity of island fauna and flora with a marked benefit for adjacent reef ecosystems[41,42]. Some predatory birds such as *Circus maillardi* also strongly participate in the regulation of small introduced mammals[43]. More generally, even at low abundance, ecologically rare mammal or bird predators may have major influence on ecosystem functioning and associated services through top-down controls along the foodweb[44]. For instance, bats, an order with a high proportion of ecologically rare species, are able to reduce arthropod abundance by 84% in agroforestry[45].

Establishing a universal list of core traits in order to measure ecological rarity in absolute terms is a difficult task given the multitude and species-specific functions that organisms can perform in ecosystems. An alternative, based on trait conservatism, would be to use species evolutionary distinctiveness as a proxy for species functional distinctiveness[10,46,47]. We found that ecological

rarity is over-represented in some orders: Primate and Chiropotera for mammals and Psittaciformes, Caprimulgiformes, and Apodiformes (Strisores) for birds. Some of these clades are well known to be highly threatened[48]. For instance, primates are highly sensitive to land-use changes with ~60% of the world's species having a high risk of extinction[48]. Yet, we found that overall ecological rarity is widely distributed across the phylogeny for both taxa, in line with the widespread distribution of ecological rarity across the functional space, and that ecologically rare species are no more evolutionarily distinct than other species. This suggests that evolutionary distinctiveness cannot be used to infer ecological status, such as ecological rarity, to prioritize global conservation strategy[18,49]. However, the relationship between ecological rarity and evolutionary distinctiveness will need to be more thoroughly examined in the future as the rates of speciation and species extinction coupled to trait diversification may blur the link between functional and evolutionary distinctiveness[11]. Ecologically rare species may be the last representatives of ecological strategies depleted by selection and provide opportunities to examine the relationships among functional traits, geographical distribution, species formation, and extinction[50].

The global distribution of ecological rarity provides complementary information when compared to geographical rarity of vertebrates[51] and their functional redundancy[26]. We identified regions that include many endemic and rare species previously highlighted by different studies[27,52,53]. Yet, we found an aggregation of ecological rarity in a handful of hotspots. Since geographical restrictiveness is a component of ecological rarity, hotspots of mammal ecological rarity are located in the major centers of endemism for Chiroptera, i.e. Southeast Asia, Taiwan, the Caribbean, the Neotropics, and Madagascar[53]. Similarly, hotspots of bird ecological rarity are spatially congruent with

endemism and the history of bird diversification[54–56]. The Andes, for instance, are well known to host a high diversity of hummingbird species adapted to high altitude[40]. Also, well known to host endemic species, islands disproportionately host ecologically rare species. This over-representation on islands could emerge from speciation and adaptation outcomes in the context of ecological opportunity produced by isolation[57]. Except for a few hotspots holding up to 12 ecologically rare mammals and 28 birds, most areas host very few ecologically rare species, making it difficult to establish a simple conservation strategy to protect ecological rarity. We also found that areas with ecologically rare species host more ecologically rare species than expected by chance, suggesting that particular environmental conditions (such as paleoclimate[54]) may shape the distribution of ecological rarity beyond the level of species richness. Besides hotspots, we also reveal the presence of ecologically rare mammals and birds in many different regions, suggesting that different combinations of paleoclimatic fluctuations, environmental variables, and plate tectonic processes[58] contribute to the emergence of ecological rarity.

Species ranges differ latitudinally. More precisely, the smallest range sizes are attained on islands, in mountainous areas, and largely in the southern hemisphere[59]. Therefore, species at high latitude have in general a lower restrictiveness than tropical species, reducing the number of ecological rare species in these areas. However, considering only functional distinctiveness we found that the northern hemisphere hosts a lower number of functionally distinct species than the southern hemisphere (see Supplementary Fig. 12). Finally, a future focus on the very few cells characterized by very high Standardized effect sizes (SES, where ecologically rare species are disproportionately over-packed) could highlight the role of ecologically rare species on ecosystem functioning. Yet, most cells do not host ecologically rare species, suggesting that ecological rarity remains a very limited phenomenon in space.

We show that ecologically rare species are disproportionately packed in IUCN threatened categories, indicating that ecological rarity is globally under threat (Fig. 5). Yet, 13% and 52% of ecologically rare mammals and birds were classified as "Not Threatened-Least Concern" and 16% and 3.8% were "Not Evaluated or Data Deficient" (such as the Ethiopian big-eared bat, *Plecotus balensis* or the coppery thorntail *Discosura letitiae*). This finding suggests to use ecologically rarity as a complementary conservation facet to identify species that are worthy of particular conservation attention because of their functional distinctiveness and possible specific role on long-term ecosystem functioning[60] even if not threatened in terms of demography or occupancy (IUCN status)[60,61].

Evidence is accumulating that ongoing climate changes are already affecting living organisms[62–64]. Taxa with limited geographic distributions are generally highly vulnerable to shifting environmental conditions and are likely to be the most affected[17,65]. Here, we show that ecologically rare species do not escape the rule. Climate change is likely to drastically reduce the geographic ranges of ecologically rare mammals and birds (Fig. 5), driving some of them towards global extinction. This concerns particularly ecologically rare birds species due to their over-representation in tropical mountains[66]. These ecosystems are classified as highly vulnerable to climate change impacts[67], mainly due to the narrow spatial distribution and environmental niche of many taxa but also to the morphological and physiological adaptations to live in such specific environments. Our predictions for future habitat suitability indicate both potential winners and losers among ecologically rare species. For instance, *Myrmecobius fasciatus* (a marsupial whose diet consists almost exclusively of termites[68]) could lose 65% of its current range while *Sylvisorex konganensis*, a small shrew, could double its range. Yet, our scenarios are based on climate change only. Habitat transformation induced by human land-use is another important driver of the ongoing mammal and bird biodiversity loss but was not considered in our models[69]. For instance, Madagascar, which hosts 26 ecologically rare mammals, has already lost 50% of forest cover between 1950 and 2005[70]. More generally, human footprint has a negative impact on restriction and fragmentation of species habitats[71] but also mammal mortality[72] and should be included in conjunction with climate change to predict the future distribution of ecological rarity. Since we show that ecologically rare species occur in areas where human footprint is higher than in areas hosting common species, they should indeed face a multiple jeopardy in the very near future[4,69,73].

Political and socio-economic factors are vital in conservation strategies and outcomes[74]. Poverty and conflicts contribute to poor governance, unsustainable bushmeat hunting, wildlife trade, and anthropization of untouched habitats, thereby potentially increasing pressure on species and their habitats[75]. Interestingly, we observe a continuum of political and socio-economic contexts across areas where ecological rarity occurs. Both countries with a low number of conflicts and a high human development index (HDI, such as Australia or the United States of America) and countries with a high number of conflicts and a low HDI (such as Indonesia or Madagascar) host ecologically rare species (see Supplementary Figs. 8 and 9). Even if countries with many conflicts and low HDI host more ecologically rare species, our study suggests a shared and transboundary responsibility for the conservation of ecological rarity between the developing and the developed world[76]. Coordinating and sharing objectives and responsibilities among countries or region may significantly increase the cost-effectiveness of conservation strategies targeted towards ecological rarity[76].

Although the percentage of the land surface devoted to protected areas has globally and markedly increased and has motivated research in conservation for decades, we show that ecologically rare species generally reach low conservation target achievement, and so are poorly covered by current protected areas. This strongly contrasts with the high level of target achievement observed for ecologically common species. Since protecting rare, threatened, or emblematic species has always guided conservation strategies[27], target achievement should be higher for ecologically rare species. Consequently to the limited resources allocated to conservation, other facets of biodiversity rather than number of species alone[27]. Given the large number of ecologically rare species currently threatened and experiencing population declines, the biosphere will soon be facing a major functional extinction crisis if effective and targeted actions are not implemented[72]. Our results urge the inclusion of ecological rarity in selecting priority conservation areas. Achieving the goal of protecting a large fraction of ecological rarity seems feasible since we reveal that a relatively small fraction of the Earth contains a large number ecologically rare species[77]. A minor but focused increase of protection on these areas could trigger large conservation gains for global diversity[27,78]. In parallel to new targets advocating the protection of 30%[79] or half of Earth[77], there is thus an urgent need and potential reward in protecting the few ecological rarity hotpots.

Humans activities are increasingly disrupting the biotic and abiotic habitat of species. We chose to work at a global scale because while a species might become locally extinct, its presence elsewhere in the region may prevent definitive extinction. Finer scales (realms, regions, or local) and thus smaller species pools could be used to refine our results and study the role of ecologically rare species. This distinction between global and local scales could be particularly important as species influence on

most ecosystem processes and services are provided locally[80]. However, we show that mammals and birds that are globally functionally distinct are also locally (inside 50 km/50 km cells) distinct (see Supplementary Fig. 13). This reflects that a species functionally different to the average of all other species existing worldwide, is also different to the average of all species it co-occurs with locally and potentially supports unique functions within its community. At the species level, we used only species geographic range to estimate geographical restrictiveness because this measure is related to global extinction risk[17,81], but other metrics such as habitat specificity and local abundance could also be used to define rarity[5,13]. In order to draw the best conservation strategies and preserve ecosystem functions, future directions will thus need to refine the measure of ecological rarity at different spatial scales and include different dimensions of taxonomic rarity. In the context of the ongoing rise in the number of functional trait databases, ecological rarity is an important divergent axis of diversity to consider in conservation while we need to evaluate more thoroughly the impact of ecologically rare species on ecosystem functioning and their contributions to humanity[12].

Our results bring insights through profiling ecologically rare species, i.e. species that are both located at the margins or in holes of the global functional space and geographically restricted. The underlying rationale is that species bearing distinct combinations of traits compared to others and filling a large volume of the functional space should be prioritized in conservation strategies owing to their unique potential contribution to ecosystem functioning. We show that the coverage of protected areas for ecological rarity was low, while human pressure and climate change could limit persistence of populations of ecological rare mammals and birds calling for a shared and transboundary responsibility for the conservation of these species.

## Methods

All analyses were done using R[82] v.3.6.0 (specific functions within specific package are indicated in italic). All relevant R code is available from the associated GitHub Repository (see section Data and Code availability).

**Distribution data.** We used the IUCN range maps[79] for 4787 terrestrial mammal species and the BirdLife range maps[83] for 9993 bird species (breeding ranges only). We removed extinct (EX) and extinct in the wild (EW) species. Given that trait and/or phylogenetic information were not available for all species, we only included species for which we had phylogenetic, functional, and distribution information (9287 bird and 4654 mammal species). Ranges were converted to 50 × 50 km equal-area grid cells. This resolution appears as a reasonable resolution to estimate local distinctiveness of each species and thus discuss the role of global distinct species at small scale. Mainland cells with >70% water were excluded but all oceanic island cells with smaller land areas were kept.

**Functional Traits.** For mammals and birds, four traits (diet, body mass (log transformed), activity cycle, and foraging height) were extracted from Elton-Traits1.0[22]. These traits are generally assumed to appropriately represent Eltonian niche dimensions of mammals or birds[22,26]. These traits have already been used to investigate community assembly rules and biogeography of both taxa[84]. We computed Gower's pairwise distances between species because we had fuzzy, categorical, and continuous traits[85]. We used the dist.ktab() function in ade4 v.1.7-6 to compute the distances[86,87]. Second, a Principal Coordinates Analysis (PCoA) identified orthogonal dimensions determining the variation in functional distances (pcoa() function in the ape package v.4.1[88]). We assessed the contribution of the traits to these dimensions and how the functional distinctiveness varied along the dimensions (see Supplementary Tables 1 and 2). Axes one to four were selected as their explained variance was greater than the null expectation of a broken stick model[89]. We then projected the species on the two axes that are the most correlated to the functional distinctiveness (PC2 and PC3 for mammals PC2 and PC4 for birds, explaining respectively 35.3% and 26.4% of the variance in the distance matrix).

**Phylogenetic data.** We used 100 randomly selected phylogenies from the posterior distribution of phylogenies from Bininda-Emonds et al.[90] with updates[91,92] for mammals, and the 100 phylogenies from Jetz et al.[93] for birds.

**Ecological rarity.** Using funrar() function in funrar v.1.3-0[15], we then computed functional distinctiveness $D_i$ of species $i$ in the global functional space[13,15], representing how original the traits of a given species are compared to all the other species from the same taxon:

$$D_i = \frac{\sum_{j=1, j \neq i}^{N} d_{ij}}{N-1} \tag{1}$$

where $d_{ij}$ the functional Gower's pairwise distance between species $i$ and $j$, $N$ the total number of species. The functional distances $d_{ij}$ are scaled between 0 and 1. $D_i$ is the average functional distance between the species of interest and all the other species of the pool. It captures how much the traits of the focal species differ, on average, compared to the rest of the species pool. $D_i$ is 0 when all species in the set have the same trait values (the functional distance between all species is 0), and 1 when species $i$ is maximally different to other species.

To test the sensitivity of distinctiveness to trait choice, each trait was deleted one at a time and distinctiveness was recomputed, then we checked for correlation between initial distinctiveness and distinctiveness with deleted traits. We did not reduce the number of traits lower because we might have missed important dimensions of the functional space defining mammals and birds niches, thus providing an over simplistic representation of the functional space (see Supplementary Fig. 14). Note that since we are using "super" trait (fuzzy trait), the depletion of one trait led to the removal of several traits (for instance depletion of Diet which is a fuzzy trait its depletion led to the depletion of 10 traits).

We also computed geographical restrictedness[13,15] using the grid cell by species matrix to measure how restricted the distribution of a species is

$$R_i = 1 - \frac{K_i}{K_{tot}} \tag{2}$$

where $R_i$ is the geographical restrictedness of species $i$, $K_i$ the number of cells where species $i$ is present and $K_{tot}$ the total number of cells. $R_i$ is close to 1 for a species present in a single cell and 0 for a species present in all cells.

Species were classified into three groups regarding their values and the quantile partitions of the bivariate space of functional distinctiveness $vs$ geographical restrictedness (Fig. 1). We defined ecologically rare and ecologically common species as having values of functional distinctiveness and geographical restrictiveness either higher than 75% or lower than 25% of the entire species pool of interest. Ecologically average species have values of functional distinctiveness and geographical restrictiveness respectively, lower than 75% and higher than 25%. With this approach we reach more or less the 5% of ecologically rare species threshold (threshold regularly used for the study of the rarity[13,94]). We performed a sensitivity analysis to test the influence of the choice of these two thresholds. We defined ecologically rare and ecologically common species as having values of functional distinctiveness and geographical restrictiveness either higher than 70% and 80%. Then, we test the correlation between the number of ecology rare species defined with the 75% (here D75R75) and these two thresholds 70%(D70R70) and 80%(D80R80). Results show that the spatial distribution of ecological rarity is robust to the choice of the distinctiveness and restrictiveness thresholds to define ecological rare species (see Supplementary Fig. 15)

To measure phylogenetic signal of ecological rarity we computed, on the 100 phylogenetic trees for each taxon, the D index[28], using the phylo.d() function in the caper v.2.0.6 package[95]. The D index is equal to 1 if ecological rarity has a phylogenetic random distribution and 0 if ecological rarity is clumped into the phylogeny. We also computed Evolutionary Distinctiveness of species[96]. The Evolutionary Distinctiveness of species i, $ED_i$, is high when the species has a long unshared branch length with all the other species. The more "isolated" a species is in a phylogenetic tree, the higher its evolutionary distinctiveness. We computed ED using the evol.distinct() function from picante v.1.6-2 package[97].

**Null model.** To account for species richness effects on the number of ecologically rare species per cell, we simulated the distribution of ecologically rare species that would be expected under the null expectation that species are randomly distributed among cells, while maintaining both species richness and frequency. For both mammals and birds, we generated 1000 random assemblage matrices using the curveball algorithm using nullmodel() function from vegan v2.4-2 packages[98]. The algorithm actually keeps the number of sites per species and the number of species per site, it only swaps occurrence patterns while keeping the marginal distributions of the site-species matrix constant. In other words, this method maintains row and column totals in a species by site matrix while shuffling presences within that matrix. As such, a widespread species that occupies many sites will still occupy as many sites in the null model. Classes of ecological rarity (rare, common, average, others) were kept constant. For each randomization, we calculated the number of ecologically rare species per cells. Standardized effect sizes (SES) were obtained by comparing the observed number of ecologically rare species and the mean and standard deviation of the null distributions.

$$SES = \frac{Observed - Mean(Null)}{sd(Null)} \tag{3}$$

Values above the null expectation indicate that the cell contains more ecologically rare species than expected given the number of species in the cell, and vice-versa.

**IUCN status**. We used the *taxize* package v.0.7.8[99] to retrieve up-to-date IUCN status for mammals and birds[79]. For easier interpretation, we grouped species in three categories: Critically Endangered (CR), Endangered (EN), and Vulnerable (VU) species as "Threatened" (TH); Least Concern (LC) and Near Threatened (NT) species as "Least Concern" (LC); Data Deficient (DF) and Not Evaluated species as "Not Evaluated" (NE). We performed a multiple comparison Kruskal-Wallis rank-sum test to compare indices across IUCN categories. We used the *kruskal()* function of the *agricolae* package v.1.2-6[100].

**Exposure to human pressure**. For each species, we assigned three indicators of exposure to human pressure. (a) Global human footprint, which measures the cumulative impact of direct human pressures (extent of built environments, crop and pasture lands, population density, night-time lights, railways, roads, and navigable water-ways) on nature between 1993 and 2009[101]. (b) Human development Index (HDI), which captures elements of life expectancy, education, and wealth for the year 2017[102]. (c) Human conflicts, which sums the years of conflicts in each country between 1946 and 2015[103]. Each dataset was rescaled at 50 km × 50 km resolution and we computed the average value over each species' spatial distribution.

**Global biodiversity scenarios**. We used a similar approach as Thuiller et al.[32] and assessed potential climate change impacts on mammal and bird species under climate change scenarios using an ensemble projection framework. More specifically, we related species distribution to four climate variables describing current climate (1979–2013) derived from the CHELSA dataset:[104] annual mean temperature, annual temperature range, annual sum of precipitation, and precipitation seasonality.

Four SDM algorithms available in the *biomod2* package[105] were used to estimate these species-climate relations: Generalized Linear Model, Generalized Additive Model, Boosting Regression Trees, and Random Forest. Models were calibrated using 80% of the initial data and evaluated against the remaining 20% of data using the True Skill Statistic (TSS[106]). Data were randomly assigned to each sub-dataset and this step was repeated four times in order to perform a robust cross-validation. Only models with a TSS > 0.8 were kept and projected into future conditions. Consequently, we kept 28% (67 species) and 59% (337 species) of ecologically rare mammals and birds, respectively. To consider realistic species dispersion, we selected absences in 3000 and 4000 km buffer around mammal and bird species range, respectively (see Thuiller et al.[32] and See supplementary material for further details).

These models were then projected under future climate change scenarios derived from five Global Circulation Models (GCMs) run under the Representative Concentration Pathway (RCP) 8.5 from the Coupled Model Intercomparison Project (CMIP5). These GCMs are the following: the CESM1-BGC[107] run by the National Center for Atmospheric Research (NCAR), the CMCC-CMS[108] run by the Centro Euro-Mediterraneo per i Cambiamenti Climatici (CMCC), the CM5A-LR[109] run by the Institut Pierre-Simon Laplace (IPSL), the MIROC5[110] run by the University of Tokyo, and the ESM-MR[111] run by Max Planck Institute for Meteorology (MPI-M). Future projections were made for two time slices: 2041–2060 (horizon 2050) and 2061–2080 (horizon 2070). For a given horizon, projections were aggregated using the weighted average consensual approach[112] using the TSS as weight.

Potential climate change impacts were assessed by comparing current and future species distribution projections using the Species Range Change (SRC[105]).

**Levels of protection and gap analysis**. To estimate the extent to which the current terrestrial protected area network covers ecologically rare species, we carried out a gap analysis following the methodology proposed in Thuiller et al.[11]. We defined a conservation target for every single species, which, in terms of distribution range within the protected areas network, represents the desired level of protection we considered necessary for a species to be adequately protected. Species-specific targets were defined based on species range sizes since restricted species require more coverage than widespread ones to avoid extinction[11,65]. Delimitation of species-specific conservation targets is the most "subjective" part of gap analysis. Specific targets are interrelated to species geographic range sizes; restricted species requiring to be more cover by protected areas than widespread ones to limit extinction risk[11,65,113]. This species-specific conservation target is the proportion of a given species geographical ranges that had to be cover by protected area network to secure their persistence. Hence, following previous works on gap analysis[11,65,113] we fixed conservation targets to be inversely proportional to log-transformed species' range sizes. We fixed that species with the smallest range needed 100% of their range to be protected, whereas widespread species only needed 10%. We fitted a linear regression between these two values to define the target for each species (see the electronic supplementary material of Thuiller et al. 2015[11]). This was carried out for the two groups separately since they harbor very different distributions of range sizes. Geographic range size data (in km²) was estimated in the Behrmann projection using species' extent of occurrence polygons from the IUCN[79]. The proportion of range currently covered by protected areas for each species was extracted from the World Database on Protected Areas (WDPA). This proportion was divided by the defined target to estimate species target

achievement; i.e., how far defined targets for each species is realized. Note that we restricted analyses to protected areas classified as strict protected areas, i.e., Ia, Ib and II by IUCN.

**Reporting summary**. Further information on research design is available in the Nature Research Reporting Summary linked to this article.

## Data availability
All data used in this paper are freely available and downloadable from the web. Species distribution maps were provided by the Mammal Red List Assessment (http://www.iucnredlist.org/). For birds, breeding range distribution maps were extracted from BirdLife (http://www.birdlife.org/). All climatic data are available on the CHELSA data portal (https://chelsa-climate.org/). IUCN status are available on the IUCN red list (https://www.iucnredlist.org/). Spatial polygons of protected areas were provided by the WDPA (https://www.protectedplanet.net/). Human development index was provided by UNDP, Human Development Indices and Indicators: 2018 Statistical Update. (2018). (http://hdr.undp.org/en/2018-update) and number of armed conflicts by Armed Conflict Dataset (2016). https://ucdp.uu.se/downloads/index.html#armedconflict. We provide for each species, coordinates on the PcoA, value of distinctiveness, restrictiveness, and all values of threats analyzed in the present paper (https://github.com/FRBCesab/ecorar).

## Code availability
All relevant R code is available from the GitHub Repository: https://github.com/FRBCesab/ecorar. R code to perform Global biodiversity scenario is available from the GitHub Repository: https://github.com/FRBCesab/free-sdm.

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

## Acknowledgements

This research is a product of the FREE group funded by the Centre for the Synthesis and Analysis of Biodiversity (CESAB) of the Foundation for Research on Biodiversity (FRB) and EDF. WT received funding from the ERA-Net BiodivERsA—Belmont Forum, with the national funder Agence Nationale pour la Recherche (ANR-18-EBI4-0009), part of the 2018 Joint call BiodivERsA-Belmont Forum call (project "FutureWeb""). CV was supported by the European Research Council (ERC) Starting Grant Project 'ecophysiological and biophysical constraints on domestication in crop plants' (grant ERC-StG-2014-639706-CONSTRAINTS).

## Author contributions

N.L. and N.M. conceived the study. W.T., C.V., D.M., and N.C. contributed to the study design. N.L., N.M., N.C., J.R, M.Gu, M.Gr, and L.V. contributed to the acquisition and analysis of data. N.L., N.M., W.T., C.V., D.M., A.O, B.M, and C.T. contributed to the interpretation of results. N.L. and N.M. wrote the first version and all authors edited the manuscript.

## Competing interests

The authors declare no competing interests.
