## [Peer Review File · Nature Communications]

Reviewers' Comments:

Reviewer #2:

Remarks to the Author:

I have now reviewed the revised version of the manuscript "Worldwide ecological rarity of mammals and birds". I commend the authors for their work addressing all comments from reviewers and revising the manuscript accordingly: this new version remains enjoyable to read and the additional analyses strengthen the conclusions further. My original concerns have been addressed satisfactorily and I have no major concerns on this revised manuscript. However, I do include a number of minor comments and edits which could be helpful to include in a final version.

Line edits (page numbers, line numbers):

Abstract

2, 35-36: If I follow the results correctly, the statement "disproportionately sensitive to current and future threats" is incorrect as applied to both ecologically rare birds and mammals, since ecologically rare mammals actually appear to be less impacted by predicted future climate change than ecologically average or common mammals.

2, 36: delete "in" after "While,".

Introduction

4, 93: Replace "these information" with "this information".

Results

5, 122: The past tense is used here ("did not overlap") while the previous clause in this sentence is in the present tense.

6, 154-161: This paragraph and associated results do not appear to add too much in my opinion and instead deflect the attention from more meaningful results. I would consider dropping most of this paragraph and relegating the SES results to the supplement entirely.

6, 161: Please clarify why this is interesting and what the implications might be as it is not immediately obvious.

6, 175-176: The statement "more impacted" here is not really supported; the analyses included indicate a spatial overlap between human footprint and the number of ecologically rare species but do not allow concluding that ecologically rare species are more/less impacted by a higher human footprint.

Discussion

8, 219: This conclusion appears a little unclear and/or circular to me. If I am not mistaken, the authors specify ecologically rare and common species as distinct by definition given that species cannot be ecologically rare if they overlap with many other species, while they cannot be common if they are far from other species. In other words, is it not unsurprising that ecologically rare and common species do not overlap in functional space since it is a consequence of the methods?

9, 254: Perhaps here or elsewhere it would be great to see a discussion (at least a mention) of the fact that ecologically rare species are possibly both ecologically distinct and geographically restricted as they may well be the last representatives of ecological strategies that are being directly selected against. Selection would make these species distinct in ecological space, if most similar species have already been pruned, and restricted in geographical space, if their ranges are contracting.

10, 283-284: Is "functionally rare" here the same as ecologically rare? If so, please be consistent.

12, 363: Remove "s" from "preserve ecosystems".

Conclusion

13, 373: "Bearing", not "baring".

Methods

14, 406: "greater than", not "greater to"

My very best,
Giovanni Rapacciuolo

Reviewer #3:

Remarks to the Author:

General comments

Overall, I think this is a very good contribution, and its revision certainly improved it. I feel that the text could still benefit from some more polishing - especially in the introduction. This section includes some overly long, convoluted and indirect sentences. Below you can find a few examples of these - but please make an effort to remedy this throughout the MS. Beyond this I have no major comments at this stage

Specific comments

Line 36 - I think that the first 'in' in this sentence is redundant

Lines 38-41 - this is a rather long and convoluted sentence, I think you can simply state that more conservation emphasis should be given to ecological rarity (no need to add the "in lieu of...").

Lines 59-62 - another long and convoluted sentence.

Lines 63-63 - here too I think you can do away with a lot of redundancy and write stronger.

Lines 65-67 - First there are a tense issues here, second your use of the word 'evaluation' twice here with two slightly different meanings does not help clarity.

Lines 67-70 - again a long, convoluted, and indirect sentence.

Line 72 – is it indeed undervalued? You begin your introduction by stating “rarity has become one of the cornerstones of many ecological research studies and conservation strategies for decade” maybe state that to-date global patterns of ecological rarity received less attention?

Line 74 – I suggest you also look at –

- Pigot et al. 2020. Macroevolutionary convergence connects morphological form to ecological function in birds. *Nature E&E* 4:230-239.
- Atwood et al. 2020. Herbivores at the highest risk of extinction among mammals, birds, and reptiles. *Science Advances* 6, eabb8458.

Line 77 – are Eltonian traits non-relevant for groups which are not birds and mammals? Please revise this.

Line 358 – probably replace ‘it’ with ‘this’

Line 373 – you probably mean ‘bearing’ rather than ‘baring’

Sup. Fig. 5 - the panels are one above the other and not besides each other as the caption would suggest.

Sup. Fig. 12,14 - you probably want to replace the word ‘line’ with ‘row’

REVIEWERS' COMMENTS:

Reviewer #2 (Remarks to the Author):

I have now reviewed the revised version of the manuscript "Worldwide ecological rarity of mammals and birds". I commend the authors for their work addressing all comments from reviewers and revising the manuscript accordingly: this new version remains enjoyable to read and the additional analyses strengthen the conclusions further. My original concerns have been addressed satisfactorily and I have no major concerns on this revised manuscript.

Thank you for these positive comments.

However, I do include a number of minor comments and edits which could be helpful to include in a final version.

Very helpful

Line edits (page numbers, line numbers):

Abstract

2, 35-36: If I follow the results correctly, the statement "disproportionately sensitive to current and future threats" is incorrect as applied to both ecologically rare birds and mammals, since ecologically rare mammals actually appear to be less impacted by predicted future climate change than ecologically average or common mammals.
We remove disproportionately and replace by "for some of them" see line 37

2, 36: delete "in" after "While,".
#Done

Introduction

4, 93: Replace "these information" with "this information".
#Done

Results

5, 122: The past tense is used here ("did not overlap") while the previous clause in this sentence is in the present tense.
#Done

6, 154-161: This paragraph and associated results do not appear to add too much in my opinion and instead deflect the attention from more meaningful results. I would consider dropping most of this paragraph and relegating the SES results to the supplement entirely.
#We keep this section because it allows to better discuss the link between the number of species and the number of ecologically rare species per cell. However, we agree that some sentences were not clear enough, and improved this paragraph. See line 151-159

6, 161: Please clarify why this is interesting and what the implications might be as it is not immediately obvious.

#We added the following sentence: *Interestingly, the number of cells with a high value of SES is very low suggesting that a few very particular environments could favor the emergence and maintenance of ecological rarity.* See Line 158-159

6, 175-176: The statement “more impacted” here is not really supported; the analyses included indicate a spatial overlap between human footprint and the number of ecologically rare species but do not allow concluding that ecologically rare species are more/less impacted by a higher human footprint.

We replaced by “*We found that geographical ranges of ecologically rare mammals and birds were respectively 1.35 ± 1 and 1.2 ± 1 times more overlapped by human footprint than ecologically common species*”. See Line

Discussion

8, 219: This conclusion appears a little unclear and/or circular to me. If I am not mistaken, the authors specify ecologically rare and common species as distinct by definition given that species cannot be ecologically rare if they overlap with many other species, while they cannot be common if they are far from other species. In other words, is it not unsurprising that ecologically rare and common species do not overlap in functional space since it is a consequence of the methods?

We agree that the previous sentences were not clear enough;

It now read (Lines 214-217): *Our findings extend this result as we find that the portion of the functional space filled by ecologically rare species does not overlap much that filled by common species, highlighting a functional complementarity, instead of redundancy, between ecologically rare and common species for both mammals and birds*

9, 254: Perhaps here or elsewhere it would be great to see a discussion (at least a mention) of the fact that ecologically rare species are possibly both ecologically distinct and geographically restricted as they may well be the last representatives of ecological strategies that are being directly selected against. Selection would make these species distinct in ecological space, if most similar species have already been pruned, and restricted in geographical space, if their ranges are contracting.

Following this comment we added this sentence: *“Ecologically rare species may be the last representatives of ecological strategies depleted by selection and provide opportunities to examine the relationships among functional traits, geographical distribution, species formation, and extinction {Ricklefs, 2005 #359}.”*

10, 283-284: Is “functionally rare” here the same as ecologically rare? If so, please be consistent.

#Done

12, 363: Remove “s” from “preserve ecosystems”.

#Done

Conclusion

13, 373: “Bearing”, not “baring”.

#Done

Methods

14, 406: “greater than”, not “greater to”

#Done

My very best,
Giovanni Rapacciuolo

Reviewer #3 (Remarks to the Author):

General comments

Overall, I think this is a very good contribution, and its revision certainly improved it.

Thank you for this positive comment.

I feel that the text could still benefit from some more polishing - especially in the introduction. This section includes some overly long, convoluted and indirect sentences. Below you can find a few examples of these – but please make an effort to remedy this throughout the MS. Beyond this I have no major comments at this stage

Specific comments

Line 36 – I think that the first ‘in’ in this sentence is redundant

#Done

Lines 38-41 – this is a rather long and convoluted sentence, I think you can simply state that more conservation emphasis should be given to ecological rarity (no need to add the “in lieu of...”).

#Done

Lines 59-62 – another long and convoluted sentence.

We reduced this sentence.

Lines 63-63 – here too I think you can do away with a lot of redundancy and write stronger.

#We removed this sentence “Geographically restricted species are usual suspects.”

Lines 65-67 – First there are a tense issues here, second your use of the word ‘evaluation’ twice here with two slightly different meanings does not help clarity.

Lines 67-70 – again a long, convoluted, and indirect sentence.

#We reduced the entire section: *Which species are facing ongoing and future threats the most? It has indeed been shown that species with restricted geographic distributions have the highest risk of extinction under most future climate scenarios. Being at risk of extinction in the face of global changes and not targeted by current*

conservation programs would be the worst-case scenario for ecologically rare species and the unique functions they support. In any case, the precautionary principle requires a sound assessment of the global patterns of ecological rarity. See line 65-70

Line 72 – is it indeed undervalued? You begin your introduction by stating “rarity has become one of the cornerstones of many ecological research studies and conservation strategies for decade” maybe state that to-date global patterns of ecological rarity received less attention?

We replace by “In any case, the precautionary principle requires a sound assessment of the global patterns of ecological rarity.” See line 69-70

Line 74 – I suggest you also look at –

- Pigot et al. 2020. Macroevolutionary convergence connects morphological form to ecological function in birds. *Nature E&E* 4:230-239.
- Atwood et al. 2020. Herbivores at the highest risk of extinction among mammals, birds, and reptiles. *Science Advances* 6, eabb8458.

#We added these two references See line 72

Line 77 – are Eltonian traits non-relevant for groups which are not birds and mammals? Please revise this.

#Done

Line 358 – probably replace ‘it’ with ‘this’

#Done

Line 373 – you probably mean ‘bearing’ rather than ‘baring’

#Done

Sup. Fig. 5 - the panels are one above the other and not besides each other as the caption would suggest.

#Done

Sup. Fig. 12,14 - you probably want to replace the word ‘line’ with ‘row’

#Done